# Modelling and Optimisation of Laser-Structured Battery Electrodes

**DOI:** 10.3390/nano12091574

**Published:** 2022-05-06

**Authors:** Lukas Schweighofer, Bernd Eschelmüller, Katja Fröhlich, Wilhelm Pfleging, Franz Pichler

**Affiliations:** 1Virtual Vehicle Research GmbH, Inffeldgasse 21a, 8010 Graz, Austria; lukas.schweighofer@v2c2.at; 2Center for Low-Emission Transport, AIT Austrian Institute of Technology GmbH, Giefinggasse 4, 1210 Vienna, Austria; bernd.eschelmueller@ait.ac.at (B.E.); katja.froehlich@ait.ac.at (K.F.); 3Institute for Applied Materials-Applied Materials Physics (IAM-AWP), Karlsruhe Institute of Technology (KIT), Hermann-von-Helmholtz-Platz 1, 76344 Eggenstein-Leopoldshafen, Germany; wilhelm.pfleging@kit.edu

**Keywords:** battery modelling, laser-structured electrodes, 3D battery concept, lithium-ion battery, multi-physics multi-domain modelling, virtual optimisation

## Abstract

An electrochemical multi-scale model framework for the simulation of arbitrarily three-dimensional structured electrodes for lithium-ion batteries is presented. For the parameterisation, the electrodes are structured via laser ablation, and the model is fit to four different, experimentally electrochemically tested cells. The parameterised model is used to optimise the parameters of three different pattern designs, namely linear, gridwise, and pinhole geometries. The simulations are performed via a finite element implementation in two and three dimensions. The presented model is well suited to depict the experimental cells, and the virtual optimisation delivers optimal geometrical parameters for different C-rates based on the respective discharge capacities. These virtually optimised cells will help in the reduction of prototyping cost and speed up production process parameterisation.

## 1. Introduction

Batteries are still the centrepiece components of electrical vehicles (EVs), which calls for their ongoing optimisation in terms of power density, energy density, cycle and calendrical lifetime, and safety. The present state-of-the-art is still formed by the lithium-ion technology. Here, a trend towards nickel-rich materials that allow a decrease of the utilisation of cobalt content dominates the cathode material development, and the transformation from pure graphite to graphite–silicon composites or pure silicon alloys determines the anode material development [1,2,3,4]. Furthermore, electrode structure optimisation is a promising pathway for improved cell performance characteristics, especially for increased high current capability.

Electrode structuring is either performed additively, similar to 3D printing (e.g., LIFT [5]) or erosively, which includes mechanical structuring (e.g., embossing [6]), chemical structuring (e.g., etching [7,8]), or laser-based methods [9,10,11].

Furthermore, methods that structure the electrodes in the synthesis process are available, e.g., the creation of interdigitated electrodes or, on an even finer level, the production of gyroid structures, which are usually applicable in special applications due to their complex structures [12,13,14,15].

Structuring of electrodes brings many advantages for batteries. In production, it can reduce the formation time of electrodes due to an enhanced wetting of electrodes with liquid electrolyte, which saves storage time and production costs [9]. Cell performance is increased due to the shortened Li-ion transport pathways through the porous electrodes, which impact the power and energy density, especially for thicker electrodes and higher C-rates [16,17]. In terms of the safety and lifespan of batteries, the improved transport characteristics in the cell lead to smaller local concentration and potential peaks, which helps avoid harmful side reactions, which can lead to accelerated ageing and thermal runaway [11].

In order to utilise the benefits of this structuring method to their full extent, a thorough understanding of their effects in the complex interaction of all involved components of a Li-ion cell sandwich is needed. This understanding can be improved and supported by computer-aided engineering (CAE) techniques. Several such CAE methods are applied by different groups to support the optimisation of electrode structuring techniques.

A simple, yet effective approach was chosen by Kraft et al. [18], where the electrode structures were condensed into improved effective conductivities in a pseudo-2D approach as developed by Newman et al. [19]. This approach keeps the simple structure of the Newman model and quantifies the overall impact of structuring, but cannot deliver insights into its role in the development of local concentration and potential gradients, minima, and maxima. Kraft et al. also published a more sophisticated approach [20], similar to the one by Chen et al. [16], where blind-hole-structured anodes were simulated with finite elements that actually resolved the shape of the hole structuring in a homogenised manner. An unhomogenised approach was presented by Latz et al. [21], where they also investigated cathode hole structuring with the holes on both electrodes set up directly opposite each other. The use of unhomogenised and separated electrolyte and active material domains supports better mapping of inhomogeneities in the electrodes at a higher computational cost. All these publications dealt with blind hole structures only. Salvadori et al. published a theoretical analysis [22] of linearly structured electrodes without a comparison to measured experimental data.

In the presented work, an electrochemical multi-scale simulation framework is developed that is capable of simulating arbitrarily structured electrodes. Both electrodes can be structured as long as a representative periodic cell can be found for the cell sandwich. The electrode 3D patterns are homogenised similar to the ones used by Chen et al. and Kraft et al. [16,18]. The model is parameterised using experimental data of line-structured electrodes.

The framework is extended by an automated meshing toolchain that allows sweeping the geometrical parameters of the structured electrodes. The impact of each parameter and their respective cross-effects on the discharge capacity at different C-rates are studied with these simulations. Specifically, the laser pitch, laser channel widths, and electrode thicknesses are varied in the frame of this publication. The results of these simulations allow for a better understanding of the influences of the varied parameters and can be used to reduce prototyping costs in production.

## 2. Materials and Methods

### 2.1. Cell Manufacturing

The pouch cells used in the discussed experiments consisted of NMC811 cathodes and graphite anodes. The cathode slurry consisted of 92% NMC811 (Targray, Kirkland, QC, Canada), 4% SUPER C65 (Imerys, Paris, France), and 4% PVDF-binder (Sigma Aldrich, St. Louis, MO, USA). The anode slurry was prepared in an aqueous formulation. It consisted of 95% artificial graphite (Hitachi, Chiyoda, Japan), 1% SUPER C45 (Imerys) and 1.5% CMC (Walocel CRT 2000 PA), and 2.5% of SBR latex (Zeon BM-451B). The electrodes were dried and compacted and used in single-layer pouch cells with a bi-cell design (1 double-sided cathode, 2 double-sided anodes) with a PP/PE/PP composite separator of 20µm (Celgard, Charlotte, NC, USA). The cells were filled with a Solvionic electrolyte (EC:EMC 3:7, 2% VC, 1M Li concentration). The areal capacity of the electrodes was $3.95\;{mAh/cm}^{2}$ and $4.7\;{mAh/cm}^{2}$ on the cathode and anode, respectively. This gives a high balancing factor of 1.19. Due to the laser-induced active material removal on the anode side, the balancing factor of the cells with structured anodes was effectively decreased, which explains the need for slightly higher balancing factors in the cells with unstructured electrodes.

The 3D structuring of electrodes was established in ambient air applying a laser micromachining system (PS450-TO, Optec s.a., Frameries, Belgium), which was equipped with an ultrafast laser radiation source (Tangerine, Amplitude Systèmes, Pessac, France). After second harmonic generation, the femtosecond laser operates at a wavelength of 515nm with a pulse length of 350–400fs ($M^{2}$ < 1.2). For the laser ablation of the electrodes, a laser pulse repetition rate of 1MHz was applied while adjusting the laser scanning speed of 1000mm/s. Laser ablation generally removes electrode material locally down to the current collector in order to achieve homogeneous and accelerated wetting of the composite electrode with the liquid electrolyte [6]. In the presented studies for laser processing of the anode and cathode materials, an average laser power of 5W was used, while according to the material-dependent ablation rates, the number of laser scans was adjusted to 15 and 7, respectively.

Altogether, four different cell configurations were built for the parameterisation of the presented model. The main difference in their configuration was established by the use of either structured or unstructured anodes and cathodes. The four cell configurations were:(a)Unstructured cell;(b)Anode structured (80µm channels with a 300µm pitch);(c)Cathode structured (60µm channels with a 300µm pitch);(d)Both electrodes structured (80µm at the anode and 60µm at the cathode, both with a 300µm pitch).

These are referred to as Configurations (a), (b), (c), and (d), as in the given list. Schematic depictions of these configurations are given by an exemplary finite element discretisation in Figure 1.

### 2.2. Electrochemical Measurements

In the cycling measurements utilised for the model parameterisation, the four cell types were discharged with a constant current until the lower cut-off voltage of 3V was reached. The discharges were followed by short pauses (300s for C/10 and 14s for the other C-rates) and eventually were charged with a constant current until the upper cut-off voltage of 4.2V, followed by a constant voltage step until a tenth of the constant current and a pause. These resulting voltage profiles were used to parameterise the model equations. Altogether, three different discharge and charge cycles at C/10, 1C, and 2C were used for all four cell types.

### 2.3. Mathematical Model

The mathematical model used in this work is based on the derivation by Pichler [23] and summarised in Table 1.

These governing equations describe the concentration of lithium-ions in the electrolyte $c_{\ell }$, the concentration of lithium-ions in the active material particles $c_{s}$, the electrolyte potential $\phi_{\ell }$, the cathode potential $\phi_{c}$, and the anode potential $\phi_{a}$. Note that $\phi_{c}$ and $\phi_{a}$ are assumed to be spatially constant over their respective electrode and only change in time. This simplification is feasible when the electrical conductivity of the solid particles (∼100S/m) is magnitudes higher than the one in the electrolyte (∼1S/m), which is assumed here. Under this assumption, the classical partial differential equation usually solved to govern the distribution of $\phi_{c}$ and $\phi_{a}$ is replaced by the constraints (Equation 11), in the case of a prescribed cell current $I_{cell}$, or (Equation 12), in the case of a prescribed cell voltage $U_{cell}$.

Equations (Equation 5)–(Equation 7) describe the transport of lithium-ions in the electrolyte domain and are applicable in one-, two-, or three-dimensional domains. For structured electrodes, we have to distinguish four different electrolyte sub-domains, namely the cathode domain $\Omega_{c}$, the anode domain $\Omega_{a}$, the separator domain $\Omega_{s}$, and the blank parts of the electrodes $\Omega_{\ell }$ that were removed by the ablation process. The cathode, anode, and separator domains are all treated as porous, but homogenised structures, expressed by their respective effective electrolyte conductivity $\kappa_{eff}=\kappa \frac{\varepsilon }{\tau }$. Here, the porosity ε and the tortuosity τ express the impact of the micro-geometry on the effective conductivity. Note that the anode, with graphite basal planes oriented parallel to the current collector, is treated as an anisotropic material due to the flat-shaped particles of graphite. This means that the conductivity in the anode electrolyte domain is actually expressed by a diagonal tensor, where each entry defines the conductivity in the respective coordinate direction. The structured domain $\Omega_{\ell }$ is treated as an ideal electrolyte space with no porosity or tortuosity hindering the lithium transport (i.e., ε=τ=1).

In the electrode domains $\Omega_{c}$ and $\Omega_{a}$, the macroscopic lithium-ion transport, described by Equations (Equation 5) and (), is coupled with the microscopic particle Equation (Equation 9), which describe the solid diffusion of lithium into the active material particles. The coupling condition is given by the Butler–Volmer Equation (Equation 13), which describes the intercalation and deintercalation of lithium. The particles are assumed to be spherically symmetric, such that a one-dimensional cut in the radial direction can be used to describe the solid diffusion process (i.e., $\Omega_{sc}=\left[0,R_{c}\right],\;\Omega_{sa}=\left[0,R_{p,a}\right],\;\Gamma_{sc}=\left\{R_{c,p}\right\}$ and $\Gamma_{sa}=\left\{R_{a}\right\}$).

Note that if the electrolyte domain is chosen to be a one-dimensional cut through the electrode sandwich, then the model would resemble the structure of the famous Newman model [19]. In the presented work, the electrolyte domains $\Omega_{c},\Omega_{s},\Omega_{\ell }$ and $\Omega_{a}$ are one-, two-, or three-dimensional, which allows for the description of structured electrodes as they are presented in the next subsection.

### 2.4. Structuring Designs

In this work, the laser-generated electrode structures were divided into three major groups: line, grid, and pinhole patterns; see Figure 2. Line and grid structures consist of parallel, continuous micro-channels. The latter includes perpendicular channels in both the x and y direction. The pinhole structures are formed by cone-shaped blind holes in the electrodes on a regular grid. For these designs, the pitch is defined by the distance of either the channel centre lines or the hole centre points. The degree of active material removal is controlled by either the channel width or the pinhole diameter.

For unstructured electrodes, the properties of particles within the electrodes are more or less independent of the in-plane position (with the exception of the boundaries of the electrodes, which should have little impact on the cell’s behaviour, thus neglected here). This is a basis for the assumption of the Newman model, which allows the usage of a representative macroscopic 1D cross-section of the cell.

However, this last assumption is not feasible for structured electrodes. In the most general cases, a three-dimensional representation of the homogenised electrodes is used to describe the electrode. Symmetries in the structures are utilised to reduce the electrode to a representative periodic unit cell, as depicted in red in Figure 2 for the discussed geometries. In the case of the linear structure, this leads to a reduction to a two-dimensional representation, in the grid case, a cuboid representation, and in the pinhole case, a prismatic representation, with a triangle base shape. These periodic cells are the geometric simulation domains of the presented model, discretised by the finite element method.

### 2.5. Parameterisation

The model parameters used in this work were partially extracted from the manufacturing process, partially taken from the literature, and partially fit to the cycling data. They are summarised in Table 2, where the respective source is indicated. Measurements of the four different manufactured cell configurations (see Section 2.1 and Section 2.2) were used to fit unknown model parameters. These cells have the same electrode and electrolyte composition, but differ in their structuring. A depiction of the two-dimensional finite element mesh used to simulate them is shown in Figure 1.

These different configurations yield different anode to cathode balancing for each cell, which is advantageous in parameter fitting. Often, only measurements based on one balancing factor are available for parameterisation. In such measurements, the impact of the anode and the cathode cannot be distinguished without enough prior knowledge. For example, the OCV curve of the cell is a superposition of the two open circuit potentials of the cathode and the anode. Furthermore, any ohmic drops are superpositions of all the overpotentials occurring in the anode, cathode, and electrolyte. The four different cell configurations give more independent information on each electrode, due to the respective shift of their lithiation windows and local current densities in comparison to the other scenarios. Therefore, it is reasonable to fit electrode-specific parameters from the available full-cell measurements.

The presented model was fit to multiple measurements simultaneously, including all four cell types and their respective C/10, 1C, and 2C discharge voltage curves. Each simulation respected the individual parameter adjustments due to the respective laser structures, but other than that, there was one set of parameters used for all simulations.

The cost function in the fitting process is defined by
(1)$$f\left(p\right)=\sum_{i=1}^{12}\frac{1}{T_{i}}\int_{t=0}^{T_{i}}{\left(u_{sim,i}\left(t;p\right)-u_{meas,i}\left(t\right)\right)}^{2}\;dt,$$
where the index *i* indicates the specific measurement index, $T_{i}$ the respective terminal time, and *p* the set of fitting parameters that is evaluated.

The fitted parameters are the diffusion coefficients $D_{s}$, the Butler–Volmer reaction rate constants $k_{BV}$, and the electrode tortuosities τ. Furthermore, the coefficients ${\hat{U}}_{0}$ and ${\hat{A}}_{k},\;k\in \left(0,\cdots ,10\right)$ of the Redlich–Kister expansion (Equation 17) used to model the cathodes open circuit potential were fit.

The individual simulations were performed via an in-house-developed finite element code (based on [24]), which is highly optimised for single-thread transient simulations of several thousand degrees of freedom. This yielded an average simulation time of around 6 s per conducted simulation. The fitting algorithm used a mix of steepest descent, random evaluations, and single parameter sweeps to minimise the cost function (Equation 1). The random evaluations and single parameter sweeps around the best parameter candidates help leaving local minima, which is a well-known problem for the steepest gradient and Newton-like optimisation methods. Due to their single-thread nature, the individual simulation could easily be performed in parallel by the python multiprocessing library pathos.

### 2.6. Initialisation

The consistent initialisation of the discussed models for the individual structured electrodes is non-trivial and will therefore be explained here in detail.

For every experiment, the initial voltage $U_{0}$ and the degree of ablation of active material $x_{c}$ and $x_{a}$ influenced the specific initialisation. The assumptions here were that initially (before formation), all the cyclable lithium $a_{Li}$ is contained in the cathode. Therefore $a_{Li}$ in mol is given by $a_{Li}=c_{max,c}m_{c}/\rho_{c}$, where $c_{max,c}$ is the maximum concentration of lithium, $m_{c}$ is the mass, and $\rho_{c}$ is the density of the cathode. A part $a_{SEI}$ of the lithium was lost in the initialisation due to the SEI formation on the anode. This part was assumed to be proportional to the size of the anode and therefore given by $a_{SEI}=x_{SEI}c_{max,a}m_{a}/\rho_{a}$, where $x_{SEI}$ is the respective proportionality factor, and $c_{max,a}$, $m_{a}$, and $\rho_{a}$ are the maximum concentration of lithium, the mass, and the density of the anode, respectively. Under these assumptions, any feasible tuple of fully relaxed lithium concentrations $\left(c_{0,c},c_{0,a}\right)$ has to fulfil the equations
(2)$$c_{0,c}m_{c}/\rho_{c}+c_{0,a}m_{a}/\rho_{a}=a_{Li}-a_{SEI}$$
and
(3)$$U_{ocv,c}\left(c_{0,c}\right)-U_{ocv,a}\left(c_{0,a}\right)=U_{0}.$$

Expressing $c_{0,c}$ from (Equation 2) and putting it into (Equation 3) yield
(4)$$U_{ocv,c}\left(\frac{\rho_{c}}{m_{c}}\left(a_{Li}-a_{SEI}-c_{0,a}m_{a}/\rho_{a}\right)\right)-U_{ocv,a}\left(c_{0,a}\right)=U_{0},$$
which was solved for $\left(c_{0,c},c_{0,a}\right)$ at the beginning of each simulation via a bi-section method. We remark that the proportionality factor $x_{SEI}$ was fit to 9.89%, which agrees with the literature values [25,26].

### 2.7. Parameter Studies

The model, which was parameterised based on the measurements acquired from the four cell types, as described in Section 2.5, was used for the simulation of all possible cell structures and their virtual optimisation. All the discussed models share the same set of electrochemical parameters and differ only in their macroscopic geometry, which describes the structured electrodes and the electrolyte space embedding them.

All three structuring methods were varied and analysed with respect to the applied laser pitch and the degree of ablation of the active material due to structuring and the electrode thicknesses (i.e., loading). This allowed for a comparison of the different structuring methods and an estimation of their respective performance gain, which was quantified by the discharge capacity of a cell configuration at different C-rates. This was chosen due to its simplicity, alongside its capability to quantify the performance at different C-rates. For most of the configurations, the discharge capacities were compared to a C/10 discharge capacity.

All the discussed studies were performed using an automated toolchain that was developed for this purpose. It consisted of a python script collection that automates the distribution of multiple simulations to a CPU cluster and the automated evaluation of their results. Each simulation then used a scripted meshing algorithm (utilizing SALOME [27]) and an in-house-developed FEM toolbox [24] that performs the simulation of the discussed cycling protocols. Altogether, about 8400 such simulations were performed to deliver the presented results. Each of these simulations included several charge and discharge cycles at different C-rates. The computational time for these simulations varied from around 6 s to around 1 h. This range stems from the range of the necessary degrees of freedom, which grew with the size of the periodic computational domain. This domain grew with the applied laser pitch distance, electrode thickness, and the dimension of the computational domain.

## 3. Results

### 3.1. Parameterisation

The parameterisation process described in Section 2.5 delivered a set of parameters that is presented as a part of the parameter summary in Table 2. A direct comparison of the measurement and fitted simulation discharge capacities is shown in Figure 3. It can be seen that the simulations of all four cell configurations over all C-rates delivered results that were very close to the measured discharge capacities.

The parameterised model was used to simulate NMC811/graphite full cells for multiple laser structure settings. There were multiple parameters/settings that could be varied and possibly influence the effects of the laser structure on the capacity.

The results of the evaluated simulations are presented and are described in the following subsections.

We remark that when the figures show C-rates for different values for the degree of ablation, then the C-rates always represent the C-rates with respect to the reference cell with unstructured electrodes. For example, consider a 2Ah cell, then 2C are equivalent to a current of 4A. If a simulation shows the capacity at 2C for a cell with an ablation of 10%, then these 2C are equivalent to 4A, albeit the 10% reduction of the active material.

### 3.2. Variation of the Degree of Ablation

First, a simple study of the impact of laser-structured electrodes is presented. For this purpose, simulations were conducted where either one or both electrodes were laser structured in a line pattern design. The laser pitch was fixed to 200 µm, which is a value commonly used in the literature [9,28]. The material loading and electrode thicknesses were the same as in the measured cells. The degree of ablation was varied from 0% to 50%. The results of the simulations where only one electrode was structured are shown in Figure 4. The figure shows the discharge capacity of full cells, where either the cathode or the anode were structured. In a second step, both electrodes were structured with an individual degree of ablation. The results are shown in Figure 5.

For these simulations, the virtual cell was charged with a constant current at C/2 until the upper cut-off voltage of 4.2V was reached, followed by a constant voltage step until the current was below C/20. The discharge steps were a constant current (C/10–3C) until the lower cut-off voltage of 3V was reached. Between each charge and discharge, the simulated cells rested for 14 s.

To show the influence of optimal structuring on the electrode utilisation, several snapshots of the lithium concentration in the electrolyte and the electrodes are shown in Figure 6. Here, the optimal structuring for a linear structuring method, at 3C discharge, is depicted at the moment where the 3C discharge cycle reaches its cut-off voltage of 3V. To be able to compare this to non-optimal structuring, also 1/2 and 1/10 of the respective degree of ablation were simulated, and each version is depicted at the moment where it reached the lower cut-off voltage. The snapshots show how the optimal structuring allowed for a deeper lithiation and de-lithiation of the anode and cathode, respectively. The broader channels at the respective optimal structuring allowed for better replenishment of lithium ions from the anode to the cathode, which only stopped when the reaction front enveloped a very small, unutilised area. The local current density caused very high local concentration gradients and an over-potential high enough to reach the cut-off voltage. In addition to the improved discharge properties, the previous charging led to higher initial lithiation and de-lithiation of the respective optimally structured electrodes. This can be seen, e.g., from the higher anode lithiation in the unutilised area. For completeness, similar snapshots are shown for the optimal grid and pinhole structures in Figure 7 and Figure 8.

### 3.3. Variation of Pitch

In the next step, also the pitch of the laser generated channels and pinholes was varied. The presented model does not describe the mechanical stability of the structured electrodes. This means that even the thinnest walls, which were created by a small laser pitch, performed ideally in a mechanical sense, and the following results have to be interpreted under the consideration of these simplifications. A consequence of this simplification is that a smaller laser pitch, in combination with thin laser channels or holes, always resulted in the best performance due to its increase of the macroscopic electrode area. As a consequence, a smaller laser pitch would always be preferable, but is only realistically advantageous as long as the mechanical stability of the electrodes can handle cycling. Still, it is of interest to study the variation, in the presented way, to quantify its impact. The model limitation, which is given by the neglected mechanical stability of the electrode structures, has to be kept in mind when interpreting these results.

The simulated discharge capacities for structure pitches between 20 µm and 300 µm and the ablation of 0–50% for line (left), grid (centre), and hole (right) structures for C-rates of C/10–3C can be seen in Figure 9. Due to the higher impact of the structuring on the anode, as can be seen in Figure 4, only anode structuring was considered for this study.

### 3.4. Variation of Electrode Loading

In the third and final virtual variation, not only the degree of ablation and the laser pitch were varied, but also the electrode loading, which is represented by the electrode thickness under the assumption of constant porosity. The results are shown in Figure 10. Here, the volumetric capacity is given instead of the absolute capacity. The specific capacity was calculated under the assumption of a 12µm anodic and a 20µm cathodic current collector foil per cell stack.

As a representative quantification of the theoretically possible performance gain, the optimal degree of ablation for the anode and its respective volumetric discharge capacity are indicated in Figure 10. Some of these results are only theoretically reachable due to the mechanical instability of the electrodes at the small optimal pitch of 20µm. From experience, values of 50µm are realisable.

The thicknesses of both electrodes were varied from 0.5- to 3-times the initial thicknesses of the reference cell electrodes of 75µm and 115µm, for the cathode and anode, respectively. For this study, similar load profiles as for the previous ones were used, where the C-rates were adjusted according to the new loading of the electrodes.

## 4. Discussion

The model resulting from the fitting process presented in Section 3.1 shows good agreement with the measured cell capacities. Therefore, it is shown that the parameterised model is capable of reproducing and predicting the structuring experiments to a satisfying degree. This allows for the virtual analysis and optimisation of electrode structuring as presented in this work.

The impact of laser-structuring either one or both electrodes, as presented in Section 3.2, shows that the initial electrode load balancing is a major factor in the difference of the individual electrode impacts. For low C-rates, structuring the cathode showed a decrease of capacity directly proportional to the degree of ablation of the active material. This is explained by its limiting role in the initial electrode balancing. For the anode, this decrease in discharge capacity was only observed when the loss of material outweighed its oversize due to the balancing. Here, the neglection of fresh SEI growth in the presented model possibly forms a model limitation that has to be considered when interpreting these results. Nevertheless, for higher C-rates, a trend emerged that favoured the anode for structuring. This can be explained by the distinction in the electrochemical differences of the electrodes, the anisotropic properties of the anode due to its flake-like-shaped particles, and the higher electrode thickness in comparison to the cathode [29].

In general, the structuring benefits increased for higher C-rates. For these, higher lithium-ion concentration gradients built up, leading to worse transport properties due to the decreased conductivity of very low and very high concentrations. The additional pathways for lithium-ion transportation due to the laser ablation reduced the resulting gradients within the electrolyte.

The definition of an optimal structuring is non-trivial and must be aligned with the final cell application and correlated performance indicators.

The simulations showed similar potential performance improvement for all three structuring methods, as can be seen in Figure 5, which would indicate that the choice of the method can be driven by other parameters, i.e., production cost and ease of application. However, the line patterns have the additional benefit of providing capillary structures, which boost the electrolyte wetting performance.

The two-dimensional parameter variation, where both electrodes are structured, emphasises the higher impact of structuring the anode as compared to the cathode.

In Figure 5, it can be seen that the cathode should be structured for high current applications, e.g., high power tools. At the top of each plot, the optimal material loss for the cathode and anode and the corresponding discharge capacity are shown. Besides, from the experimental data, we can conclude that the structuring of cathodes is also beneficial to reduce cell polarisation when high mass loading is applied [10,30].

The variation of the loading and its volumetric discharge performance for the structured anodes are presented in Figure 10. First of all, for C-rates up to 1C, it can be beneficial to increase the cell loading of around $4\;{mAh/cm}^{2}$. For the low C/10 discharge, the best performance is shown by the cell loading of a factor of 3. For 1C, a load increase of a factor of 1.5 would still be beneficial. For higher C-rates, the increased local current densities neutralise these beneficial effects, where the 3C discharge capacity would be highest for the lower cell loading of a factor of 0.5.

For high C-rates and thick film electrodes, an interesting effect occurred. The best values, as can be seen in the lower right plots of Figure 10, were achieved with no structuring here. This indicates that the beneficial effects of the structuring are additionally counteracted by other effects, namely the loss of the macroscopic interface of the cathode to the separator. This surface forms the initial reaction front for the intercalation and de-intercalation of lithium. Due to the high local current densities for those high loaded cells, the reaction front cannot really go much deeper without reaching the cutoff voltage, and so, any loss in the electrode to separator interfaces directly decreases the possible discharge capacity. Therefore, the unstructured cell, where the surface is still intact, performed better than the structured versions. We remark here that the discharge capacities were very low anyway due to these effects. Overall, the implemented active materials, their natural limitations, and general characteristics should be taken into account and considered.

## 5. Conclusions

An electrochemical multi-scale model, capable of simulating electrode structuring effects, was presented. It was fit to four different cell configurations that were acquired by four different structuring scenarios, all based on the same initial cell setup. The resulting parameterised model was then used to virtually optimise the cell structuring parameters degree of ablation of the active material, the structural pitch distance, and the electrode thickness based on the discharge capacities for different C-rates as a performance measure.

The quality of the parameterised model showed that the approach is fully capable of mapping electrode structuring to its impact on cell performance.

The virtual optimisation of the cells showed that all three discussed methods have roughly the same capability, but might differ in cost or production considerations, which were not discussed here and which would go beyond the scope of this work. For more details, refer to [31].

The presented model could be further extended in terms of mechanical aspects to also predict the instabilities that arise at small laser pitches. Furthermore, a full simulation of the SEI formation would allow the prediction of the long-term effects of the structuring.

## 6. Tables with Captions

The summarised model equations are presented in Table Section 6 and the respective parameters are presented in Table 1. The source of each parameter or the formula used to calculate the parameter from other parameters is indicated in the table.

**Table 1 nanomaterials-12-01574-t001:** Summary of model equations.

Electrolyte transport in homogenised electrodes:
(5) $$\begin{array}{cc} \varepsilon \frac{\partial c_{\ell }}{\partial t}-\nabla \cdot \left(\frac{RTt_{+}}{F^{2}}\frac{\kappa \varepsilon }{\tau c_{\ell }}\nabla c_{\ell }+\frac{t_{+}\kappa \varepsilon }{F\tau }\nabla \phi_{\ell }\right) & =A_{i}j_{BV},\;\;\mathrm{in}\;\Omega_{c}\cup \Omega_{a} \end{array}$$ (6) $$\begin{array}{cc} -\nabla \cdot \left(\frac{RTt_{+}}{F}\left(2t_{+}-1\right)\frac{\kappa \varepsilon }{\tau c_{\ell }}\nabla c_{\ell }+\frac{\kappa \varepsilon }{\tau }\nabla \phi_{\ell }\right) & =FA_{i}j_{BV},\;\;\mathrm{in}\;\Omega_{c}\cup \Omega_{a} \end{array}$$
Electrolyte transport in separator and structured areas:
(7) $$\begin{array}{cc} \varepsilon \frac{\partial c_{\ell }}{\partial t}-\nabla \cdot \left(\frac{RTt_{+}}{F^{2}}\frac{\kappa \varepsilon }{\tau c_{\ell }}\nabla c_{\ell }+\frac{t_{+}\kappa \varepsilon }{F\tau }\nabla \phi_{\ell }\right) & =0,\;\;\mathrm{in}\;\Omega_{s}\cup \Omega_{\ell }, \end{array}$$ (8) $$\begin{array}{cc} -\nabla \cdot \left(\frac{RTt_{+}}{F}\left(2t_{+}-1\right)\frac{\kappa \varepsilon }{\tau c_{\ell }}\nabla c_{\ell }+\frac{\kappa \varepsilon }{\tau }\nabla \phi_{\ell }\right) & =0,\;\;\mathrm{in}\;\Omega_{s}\cup \Omega_{\ell }, \end{array}$$
Active material lithium diffusion:
(9) $$\begin{array}{c} \frac{\partial c_{s}}{\partial t}-\frac{1}{r^{2}}\nabla \cdot \left(r^{2}D_{s}\right)\nabla c_{s}=0\;\;\mathrm{in}\;\Omega_{sx}\;\mathrm{for}\;x\in \left\{c,a\right\}, \end{array}$$
Lithium intercalation boundary condition:
(10) $$\begin{array}{c} -D_{s}\nabla c_{s}\cdot \vec{n}=j_{BV},\;\;\mathrm{on}\;\Gamma_{sx}\;\mathrm{for}\;x\in \left\{c,a\right\}, \end{array}$$
Charge conservation conditions when the current $I_{cell}$ is prescribed:
(11) $$\begin{array}{cccc} \int_{\Omega_{c}}FA_{i}j_{BV} & =I_{cell}, & \;\int_{\Omega_{a}}FA_{i}j_{BV} & =-I_{cell} \end{array}$$
Charge conservation conditions when the voltage $U_{cell}$ is prescribed:
(12) $$\begin{array}{cccc} \phi_{c}-\phi_{a} & =U_{cell}, & \;\int_{\Omega_{c}}FA_{i}j_{BV} & =-\int_{\Omega_{a}}FA_{i}j_{BV} \end{array}$$
Butler–Volmer reaction kinetics at the particle interface:
(13) $$\begin{array}{cc} j_{BV} & =i_{0}\left(\frac{c_{\ell }}{c_{\ell 0}}\exp\left(\frac{\alpha F}{RT}\eta \right)-\exp\left(\frac{-(1-\alpha )F}{RT}\eta \right)\right) \end{array}$$ (14) $$\begin{array}{cc} i_{0} & =k_{BV}\exp\left(\frac{F}{RT}\left(\left(\xi -\alpha \right)U_{OCP}-\int_{0}^{\xi }U_{OCP}\left(x\right)\;dx\right)\right). \end{array}$$ (15) $$\begin{array}{cc} \eta & =\phi_{s}-\phi_{\ell }-U_{OCP}, \end{array}$$ (16) $$\begin{array}{cc} \xi & =c_{x}/c_{x,max} \end{array}$$
Open circuit potential expressed by the Redlich–Kister expansion:
(17) $$\begin{array}{cc} U_{OCP}\left(\xi \right) & =F{\hat{U}}_{0}+\frac{RT}{F}\ln\left(\frac{1-\xi }{\xi }\right)+\frac{RT}{F}\sum_{k=0}^{1}0{\hat{A}}_{k}\cdot \left({\left(2\xi -1\right)}^{k+1}-\frac{2\xi k(1-\xi )}{{\left(2\xi -1\right)}^{1-k}}\right) \end{array}$$

**Table 2 nanomaterials-12-01574-t002:** Summary of the model parameter values or formulas and an indication of their respective source.

	Parameter	Anode	Value Separator	Cathode	Unit
$A_{i}$	inner surface area	$3\cdot \varepsilon /R_{p}$		$3\cdot \varepsilon /R_{p}$	$m^{2}/m^{3}$
$c_{max}$	maximal concentration	$50.055\times 10^{3}$		$31.36\times 10^{3}$	$\mathrm{mol}/m^{3}$
$D_{s}$	solid diffusivity	$3.28\times 10^{-12}$		$1.16\times 10^{-13}$	$m^{2}/s$
$k_{BV}$	reaction rate constant	$1.475\times 10^{-5}$		$2.634\times 10^{-4}$	$\mathrm{mol}/(m^{2}s)$
*l*	thickness	115 ^(2)^	25 [32]	75 ^(2)^	μm
$l_{cc}$	current collector thickness ^(4)^	12		20	μm
$R_{p}$	particle radius	19 [33]		10 ^(4)^	μm
α	transfer coefficient [23]	0.5		0.5	-
ε	electrode porosity	0.336595 ^(3)^	0.39 [32]	0.413318 ^(3)^	-
τ	tortuosity (through-plane)	4.008	1.268	1.289	-
	tortuosity (in-plane)	3.682	1.268	1.289	-
Redlich–Kister parameters ^(1)^					
${\hat{U}}_{0}$		−1.7203		3.9874995	V
${\hat{A}}_{0}$		$-0.35799\times 10^{6}$		$-6.113\times 10^{4}$	-
${\hat{A}}_{1}$		$-0.35008\times 10^{6}$		− $5.540\times 10^{3}$	-
${\hat{A}}_{2}$		$-0.35247\times 10^{6}$		− $4.526\times 10^{3}$	-
${\hat{A}}_{3}$		$-0.35692\times 10^{6}$		− $1.325\times 10^{3}$	-
${\hat{A}}_{4}$		$-0.38633\times 10^{6}$		$-2.740\times 10^{4}$	-
${\hat{A}}_{5}$		$-0.35908\times 10^{6}$		− $1.894\times 10^{4}$	-
${\hat{A}}_{6}$		$-0.28794\times 10^{6}$		$-7.237\times 10^{4}$	-
${\hat{A}}_{7}$		$-0.14979\times 10^{6}$		$-3.182\times 10^{4}$	-
${\hat{A}}_{8}$		$-0.39912\times 10^{6}$		− $8.918\times 10^{4}$	-
${\hat{A}}_{9}$		$-0.96172\times 10^{6}$		− $8.527\times 10^{3}$	-
${\hat{A}}_{10}$		$-0.63262\times 10^{6}$		− $8.527\times 10^{3}$	-
Domain independent:					
κ	electrolyte conductivity [23]	$\frac{c_{\ell }}{4.93\times 10^{8}+1.27\times 10^{9}e^{\left(9.85\times 10^{-4}c_{\ell }\right)}}\frac{F^{2}}{RT}$			S/m
$c_{\ell 0}$	initial salt concentration ^(2)^	1000			${\mathrm{mol}/m}^{3}$
F	Faraday constant	96,485.33			As/mol
R	universal gas constant	8.314			J/(kgmol)
T	absolute temperature	298.15			K
$t_{+}$	transference number [23]	0.33			−

^(1)^ All Redlich–Kister parameters for the anode come from [34]. All other values are fit. ^(2)^ Measured values.
^(3)^ Calculated values. ^(4)^ Estimated values.

## Figures and Tables

**Figure 1 nanomaterials-12-01574-f001:**
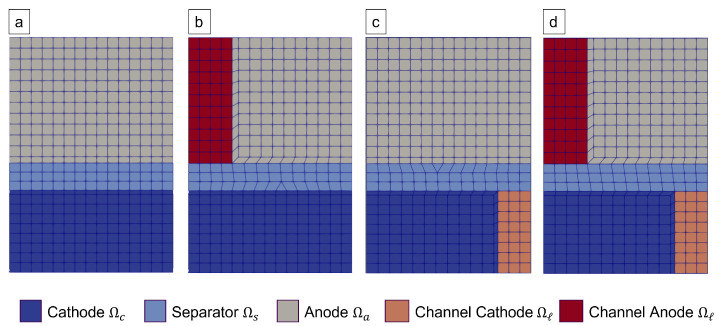
Periodic 2D finite element discretisation of the four different cell configurations used for fitting. We remark that Configuration (**a**) represents unstructured electrodes, which are equivalently simulated by a 1D cut as in the known Newman configuration. (**b**–**d**) represent the used meshes corresponding to the cell configurations as described in Section 2.1.

**Figure 2 nanomaterials-12-01574-f002:**
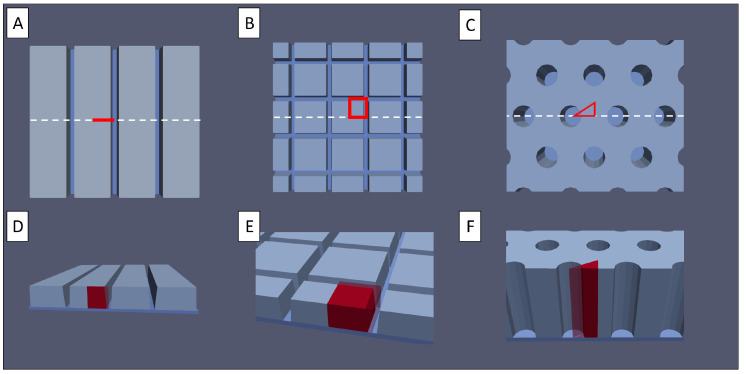
(**A**–**C**) Top view of (**A**) line structured, (**B**) grid structured, and (**C**) pinhole structured electrodes. The red marked area depicts a possible representative cross-section of the electrode. (**D**–**F**) Bird’s eye view of the same structures.

**Figure 3 nanomaterials-12-01574-f003:**
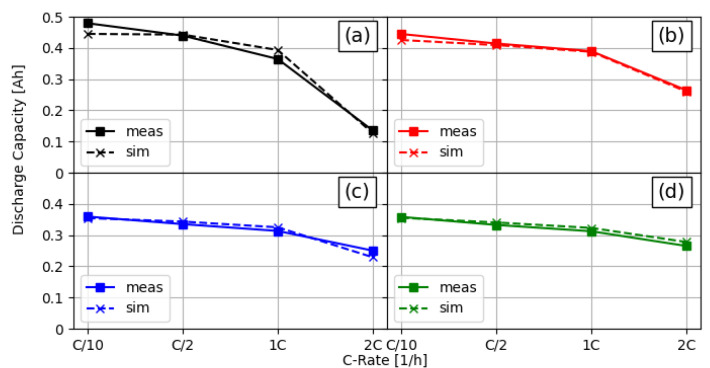
Comparison of measured and simulated discharge capacities for different C-rates and linear structured cell setups. (**a**) Unstructured reference cell, (**b**) structured anode (80µm channels and 300µm pitch), (**c**) structured cathode (60µm channels and 300µm pitch), and (**d**) both electrodes structured (80µm anode and 60µm cathode channels, both with 300µm pitch).

**Figure 4 nanomaterials-12-01574-f004:**
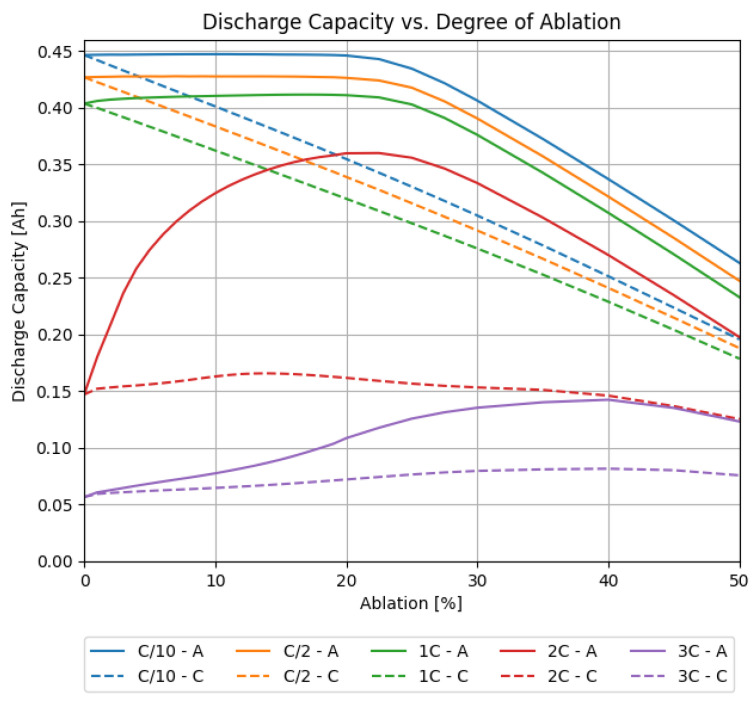
Simulated discharge capacities for full cells for different C-rates, versus the degree of ablation, where (solid) only the anode or (dashed) only the cathode was structured.

**Figure 5 nanomaterials-12-01574-f005:**
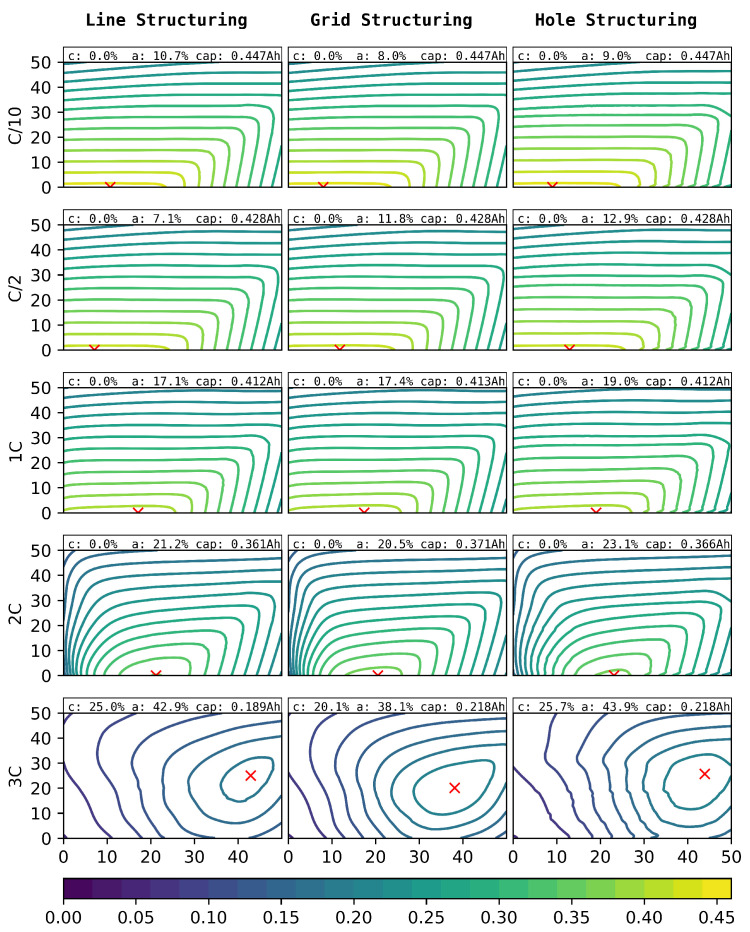
Discharge capacities [Ah] for full cells with line (left, pitch = 200µm), grid (centre, pitch = 200µm), and hole (right, pitch = 70µm) structures, for C-rates C/10–3C. The x-axis shows the degree of ablation as a percentage of the anode, the y-axis for the cathode. At the top of each plot, the optimal degree of ablation for the cathode and anode and the corresponding discharge capacity are shown.

**Figure 6 nanomaterials-12-01574-f006:**
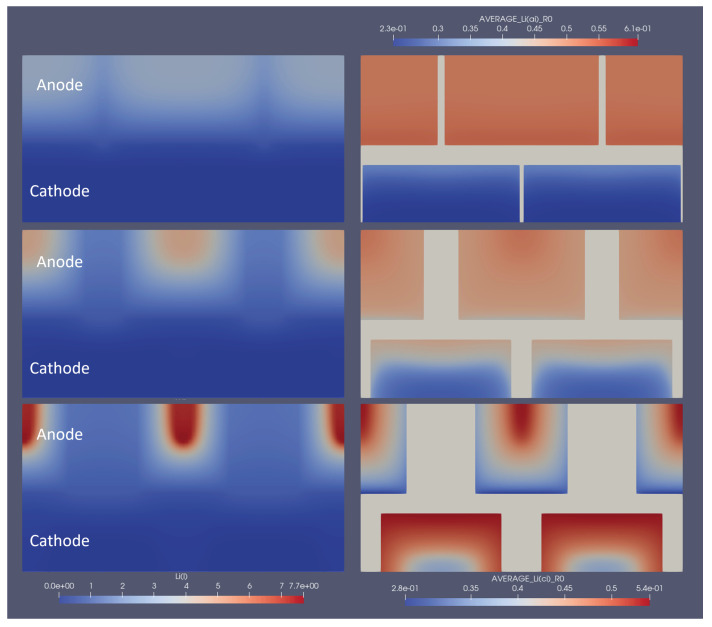
Snapshot of the lithium concentrations in electrolytes and solids for line structuring at the end of the 3C discharge cycle. The left part shows the concentration in the electrolyte, and the right part shows the average particle concentration in the respective electrodes. The bottom shows the optimal parameters, whereas the middle row shows 50% of that structuring amount, and the top part shows 10% for comparison. 2.3e-01 represents 2.3 × 10${}^{-1}$, the same rule applies to other E notations.

**Figure 7 nanomaterials-12-01574-f007:**
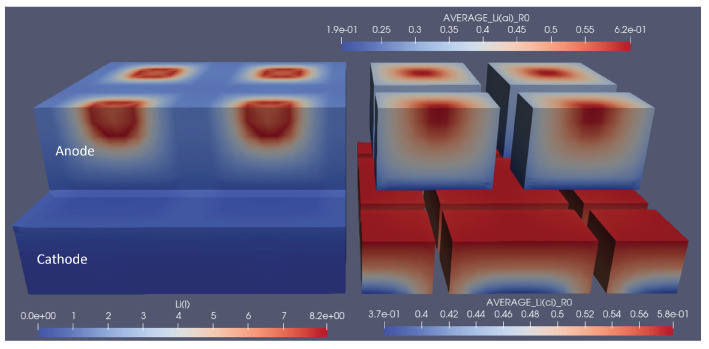
Snapshot of the lithium-ion concentrations in electrolytes and solids for the optimal grid structuring at the end of the 3C discharge cycle. The left part shows the concentration in the electrolyte, and the right part shows the average particle concentration in the respective electrodes. 1.9e-01 represents 1.9 × 10${}^{-1}$, the same rule applies to other E notations.

**Figure 8 nanomaterials-12-01574-f008:**
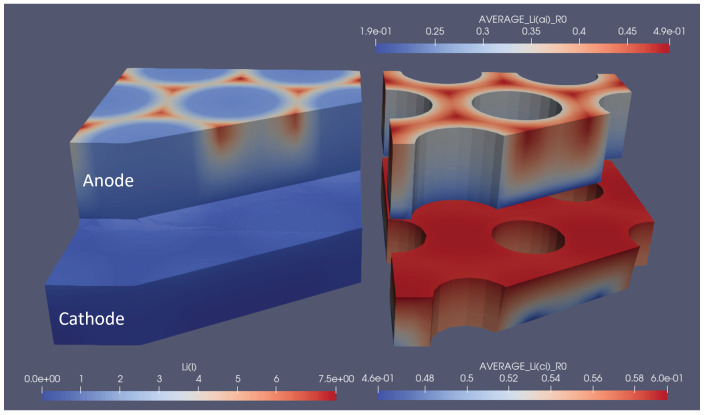
Snapshot of the lithium concentrations in electrolytes and solids for the optimal pinhole structuring at the end of the 3C discharge cycle. The left part shows the concentration in the electrolyte, and the right part shows the average particle concentration in the respective electrodes. 1.9e-01 represents 1.9 × 10${}^{-1}$, the same rule applies to other E notations.

**Figure 9 nanomaterials-12-01574-f009:**
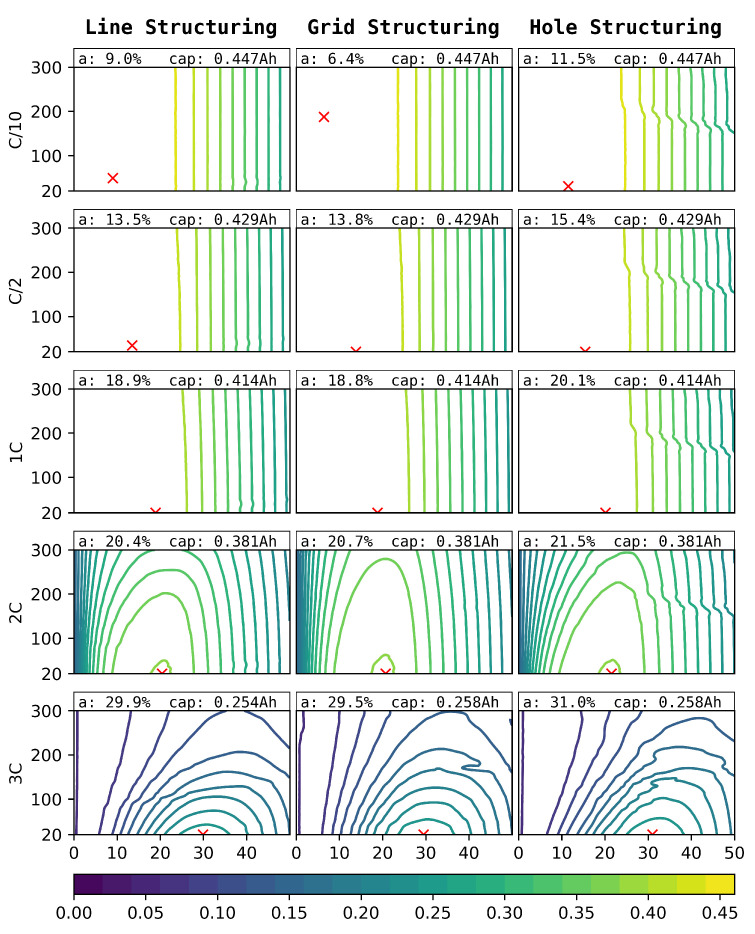
Simulated discharge capacities (Ah) for different C-rates (outer y-axis), laser pitches ((µm), inner y-axis), structure types (outer x-axis), and degrees of ablation of the anode ((%), inner x-axis).

**Figure 10 nanomaterials-12-01574-f010:**
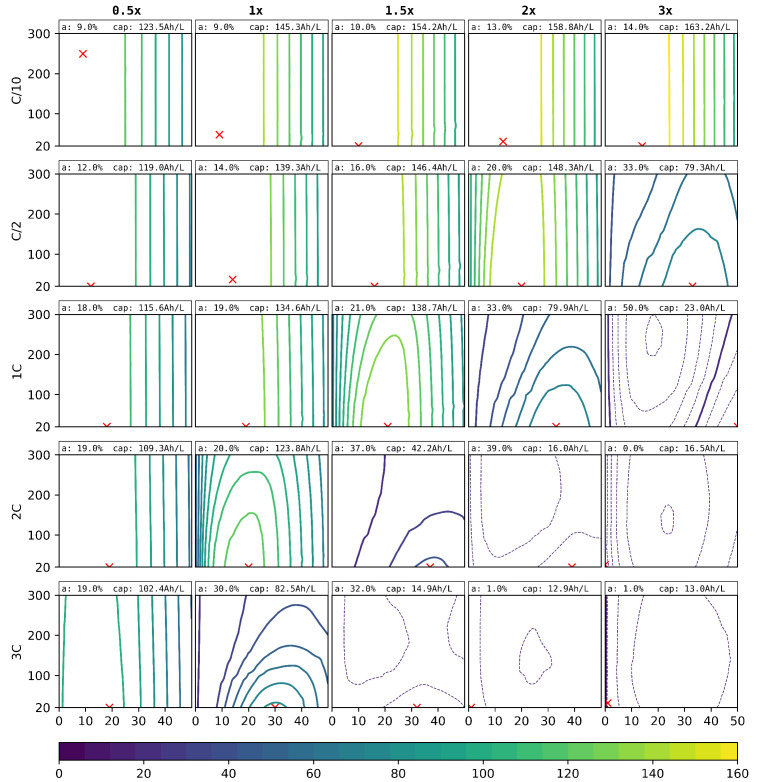
Simulated discharge capacity (Ah) for different C-rates (outer y-axis), laser pitches ((µm), inner y-axis), thicknesses (outer x-axis), and degrees of ablation ((%), inner x-axis). The red crosses show the optimal degrees of ablation and pitch for each C-rate and structuring method. The optimal parameters and the respective discharge capacity are indicated at the top of each graph. Solid lines differ by 10Ah/L and dashed lines by 1Ah/L.

## Data Availability

Data available on request due to restrictions.

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
