# Peer review of "Modelling and Optimisation of Laser-Structured Battery Electrodes"

_nanomaterials, 2022, doi:10.3390/nano12091574_

Round 1
Reviewer 1 Report
These studies demonstrated the modeling and simulation of LIBs with structured electrodes by laser ablation. The authors used their own parallel FEM code for the parameter studies with various structured electrodes. The manuscript is well organized, however, the authors must address a few issues.
1. The parameters used in this study were fitted by experimental C-V curves. However, the optimization of multiple parameters is too difficult to use in P2D models. How many errors (or cost function) were the C-V profiles observed?
2. Figure 4 shows the effects of ablation on each electrode. Under 1C condition, discharge capacity linearly decreases in the case of structured cathode electrodes. This implies that structural effects were not detected. However, the case of structured anode electrodes was not. Can the authors explain it?
3. Figure 5 (and figure 9&10) indicates relative capacity. To clarify, a scale bar is needed in figure 5.
4. Although the authors address the various structuring of electrodes for LIBs, the three different types of structuring show similar discharge capacity in terms of D.O.A, line spacing and etc. Authors must address details of the impact of this work.
5. In multiscale point of views, many other research groups demonstrated structuring electrode in porous electrode scale (patterning), secondary particles scale (3D gyroid structure) and so on. Please refer these state of art works.
Author Response
- The parameters used in this study were fitted by experimental C-V curves. However, the optimization of multiple parameters is too difficult to use in P2D models. How many errors (or cost function) were the C-V profiles observed?
Response: The following parameters have been fitted
|
Tortuosity |
|
Solid Diffusivity |
|
Kbv |
|
Redlich--Kister parameter |
As we have stated in chapter 2.5, while there were quite some parameters to fit, it was very helpful to have C-V profiles for C-rates ranging from 0.1C to 2C for different structured cells. This variance in crate and balancing between anode and cathode allows us to fit the ocv curve and the other fitting parameters directly from the measurements. The quality of the fit is quantified by the discharge capacities shown in Figure 3, where these discharge capacities are the main quantification value for all further studies in the presented work. This is why we believe that the fit is feasible and satisfying in the scope of the presented work.
- Figure 4 shows the effects of ablation on each electrode. Under 1C condition, discharge capacity linearly decreases in the case of structured cathode electrodes. This implies that structural effects were not detected. However, the case of structured anode electrodes was not. Can the authors explain it?
Response: In frame of our simulation, for 1C and below, both types of electrodes do not really show beneficial behavior. However, since the anode is not the restrictive electrode in terms of discharge capacity (balancing factor ~1.19), a small reduction in active material through ablation of the anode does not immediately yield a decreased discharge capacity on the cell level. This issue is discussed in line 337 to 342 in the initial proposal (2nd paragraph of chapter 4)
- Figure 5 (and figure 9&10) indicates relative capacity. To clarify, a scale bar is needed in figure 5.
Response: We have updated the figures with respect to your comments and fully agree that they lacked transparency which we sorted out by the update.
- Although the authors address the various structuring of electrodes for LIBs, the three different types of structuring show similar discharge capacity in terms of D.O.A, line spacing and etc. Authors must address details of the impact of this work.
Response: See line 351ff: the different structuring methods yield similar results for the observed scenario. Thus, the choice of method might be driven by other parameters/restrictions. e.g. capillary structures of linear patterns might improve wetting properties. Grid patterns might suppress mechanical degradation due to active material volume expansion. Hole patternings enable other types of laser structuring strategies, e.g. the thermal impact of to the active material by using “long laser pulses” might be less critical. However, each type of pattern needs a further process development regarding process up-scaling including special designs for optical beam shaping (multi-beam processing) and guiding as well as laser power sources with high average laser power and high repetiton rates.
These examples however are not in the scope of the publication.
- In multiscale point of views, many other research groups demonstrated structuring electrode in porous electrode scale (patterning), secondary particles scale (3D gyroid structure) and so on. Please refer these state of art works.
Response: As per your request we have updated the state of the art, to introduce the gyroid structures and particle structuring. We would like to emphasize that due to it's fundamental difference to electrode structuring, the particle structuring was not considered as highest priority in the state of the art for this work.
Reviewer 2 Report
This manuscript describes a multi-scale simulation framework to simulate and optimize various parameters of laser structured battery electrodes. The authors provided detailed analysis and rich data for their simulation framework and result. However, the manuscript may benefit from a few improvements before being published.
- The model has a large number of parameters, and it is likely to overfit with a small number of calibration cells. The authors should have some metrics to show that their model can predict unseen cells well, e.g. fabricate a few more cells and test the model’s performance on those cells.
- Figure 5 shows that faster charging-discharging favors higher degree of ablation. Can the authors discuss the implications of this result and make recommendations on how to reconcile this observation in practice?
- The authors noted that the thinnest pitch of 20 um may not be practical. Can they provide a back-of-the-envelope estimate of the minimum required thickness for electrode walls to be mechanically stable, and set that as a constraint in their optimization? This would make the results more meaningful.
- Can the authors elaborate why ablation on the anode has higher impact than on the cathode? They should discuss in more detail for line 346-348 or cite references, to better explain why higher C-rates favors the anode for structuring.
Author Response
- The model has a large number of parameters, and it is likely to overfit with a small number of calibration cells. The authors should have some metrics to show that their model can predict unseen cells well, e.g. fabricate a few more cells and test the model’s performance on those cells.
Response: The reviewer is correct, further experiments to verify the models prediction capabilities of different cell behaviour are ongoing but beyond the main scope of the present publication. In this paper, we want to present the multi-scale framework and its implementation and initial parameterization. And we believe that the 4 variations in structuring already provide a good fundament for the proposed parameterization. As we have argued in section 2.5 the different balancing of each cell composition gives us the possibility to fit these parameters without overfitting.
- Figure 5 shows that faster charging-discharging favors higher degree of ablation. Can the authors discuss the implications of this result and make recommendations on how to reconcile this observation in practice?
Added: In general, the structuring benefits are increased for higher C-rates. For these, higher lithium-ion concentration gradients are built up, leading to worse transport properties due to the decreased conductivity of very low and very high concentrations. The additional pathways for lithium-ion transportation due to the laser ablation reduce the resulting gradients within the electrolyte.
We have updated this discussion in the paper in the Discussion chapter.
- The authors noted that the thinnest pitch of 20 um may not be practical. Can they provide a back-of-the-envelope estimate of the minimum required thickness for electrode walls to be mechanically stable, and set that as a constraint in their optimization? This would make the results more meaningful.
Response: From experience, the minimum thickness for the electrode walls can be estimated to be around 50µm. However, this value is dependent on several things, such as the material, the electrode thickness, the required accuracy of the whole structuring process, and more. We have added this estimation into the paper.
- Can the authors elaborate why ablation on the anode has higher impact than on the cathode? They should discuss in more detail for line 346-348 or cite references, to better explain why higher C-rates favors the anode for structuring.
Added further explanation and citation. In the discussion section.
Round 2
Reviewer 1 Report
Authors addressed all question. I think it is worth to be published Nanomaterials.